# Constructing Variable-depth ViTs from Repeatable Low-bit Learngene

## Abstract

Large-scale Vision Transformers (ViTs) have achieved remarkable success across a wide range of computer vision tasks. However, fine-tuning and deploying them in diverse real-world scenarios remains challenging, as resource constraints demand models of different scales. The recently proposed Learngene paradigm mitigates this issue by extracting compact, transferable modules from well-trained ancestor models to initialize variable-scaled descendant models. Yet, existing Learngene methods mainly treat learngenes as initialization modules for descendant models, without addressing how to construct these models more efficiently. In this work, we rethink the Learngene methodology from the perspectives of quantization and parameter repetition. We introduce Repeatable Low-bit Learngene (RELL), which compresses ancestor knowledge into a small set of quantized, cross-layer shared modules via quantization-aware training and knowledge distillation. These repeatable low-bit modules enable flexible construction of descendant models with varying depths through parameter replication, while requiring only lightweight adapter tuning for downstream adaptation. Extensive experiments demonstrate that RELL achieves superior parameter efficiency and competitive or better performance compared with existing Learngene methods.

## 1 Introduction

Large-scale vision transformers (ViT) have achieved groundbreaking progress in various computer vision tasks (Dosovitskiy et al., 2020; Carion et al., 2020; Zhu et al., 2023; Xia et al., 2024a). However, in practical applications, models of different scales often need to be deployed and trained under diverse resource constraints—ranging from edge devices with limited storage to cluster environments with abundant resources. Simply fine-tuning a large, well-trained model (e.g., DeiT) struggles to flexibly adapt the model's scale and architecture to meet practical requirements (such as computational resources, latency, and storage limitations), as shown in Fig.1(a). Meanwhile, training each target model from scratch for different scenarios would increase training costs and compromise final model quality. Thus, a challenging question arises: How can we efficiently construct appropriately scaled models tailored to the needs and constraints of different scenarios?

To address this challenge, the Learngene paradigm (Wang et al., 2022) offers a novel approach: it compresses the knowledge of a large, well-trained model (called ancestor model) into a compact yet generalizable module called learngene. This module can be transferred to various downstream scenarios and expanded into downstream model (called descendant model) of different scales through minimal training, balancing both transfer efficiency and flexibility. Several works (Wang et al., 2022; 2023) initialize descendant models by selecting certain layers from the ancestor model as learngene and stacking them with randomly initialized layers for training. Other approaches (Xia et al., 2024c; Lin et al., 2024) employ knowledge distillation to compress the ancestor model's knowledge into several modules, which are then linearly expanded to initialize depth-variable descendant models.

However, existing Learngene methods mainly treat learngenes as initialization modules for descendant models, without addressing how to construct these models more efficiently, as illustrated in Fig. 1(b). In practice, these approaches require expanding the compact learngene back into full-scale descendants and performing full-parameter fine-tuning, which results in considerable training costs. Moreover, the learngenes are typically retained in full precision, limiting compression ratios and thereby preventing efficient inheritance of knowledge from large-scale ancestor models. Such

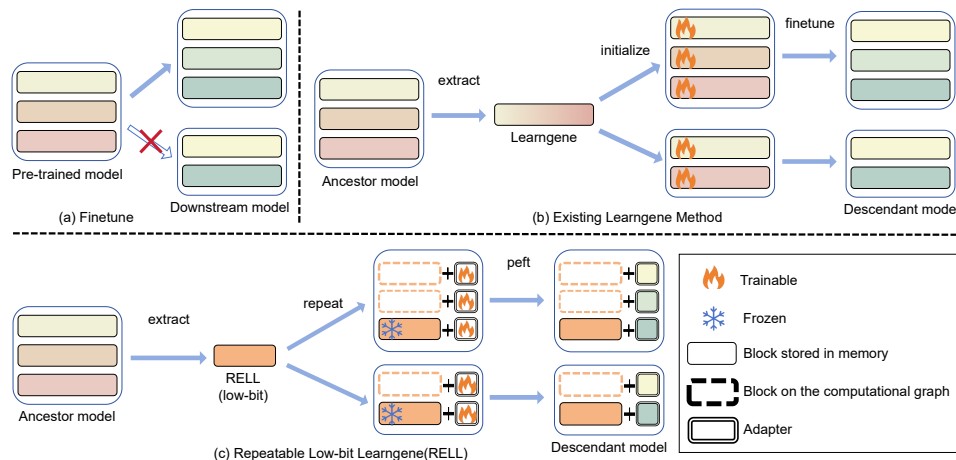

Figure 1: (a) Fine-tuning struggles to flexibly adjust model scale according to downstream task requirements. (b) Existing Learngene methods can initialize models of varying scales but require full-parameter fine-tuning. (c) RELL only needs to fine-tune adapters and further reduces model size through weight repetition and weight quantization.

limitations restrict the scalability of the Learngene paradigm and hinder its ability to fully exploit the potential of powerful ancestor models in resource-constrained scenarios. To overcome these challenges, we look beyond simple parameter initialization and explore new directions for descendant construction.

Our motivation is guided by two complementary perspectives. On the one hand, several studies (AskariHemmat et al., 2022; Zhang et al., 2023) have shown that quantization-aware training (QAT) not only compresses model parameters but also introduces a beneficial regularization effect: the clipping and discretization operations inject perturbations that guide optimization toward flatter minima, thereby reducing sensitivity to parameter noise and improving generalization. This dual benefit makes quantization inherently compatible with the Learngene framework. On the other hand, studies on parameter sharing in large Transformer architectures (Lan et al., 2019; Zhang et al., 2022) reveal that reusing the same set of parameters across layers can effectively capture cross-layer commonalities, enabling both substantial parameter reduction and flexible depth adaptation by adjusting repetition counts. Together, these insights suggest that descendant models can be built from a compact set of low-bit, repeatable modules that simultaneously achieve compression, robustness, and scalability.

Guided by these two insights, we propose the **Repeatable Low-bit Learngene (RELL)** framework, illustrated in Fig. 1(c). The central idea of RELL is to transform the knowledge of a large ancestor model into a compact yet versatile set of modules that are both quantized and repeatable. To achieve this, we first construct an auxiliary model trained with quantization-aware training (QAT) and knowledge distillation, compressing the parameters of the ancestor model into low-bit representations (e.g., 4-bit). Instead of preserving layer-specific weights, these parameters are reorganized into a small number of shared modules that capture cross-layer regularities, allowing them to function as universal knowledge carriers across different depths.

These repeatable low-bit modules then serve as the building blocks for descendant construction. By adjusting the repetition count, we can flexibly generate models of varying depths and computational scales, tailored to heterogeneous deployment scenarios. During downstream adaptation, the modules remain frozen to preserve compressed knowledge, while only lightweight trainable components—such as low-rank adapters (LoRA)—are inserted for task-specific fine-tuning. This modular design decouples knowledge inheritance from adaptation, substantially reducing training cost and storage overhead. In essence, RELL provides a unified framework that integrates model compression, modular repetition, and efficient adaptation, enabling the scalable construction of resource-efficient descendant models without sacrificing performance.

Our main contributions can be summarized as follows:

- We propose an innovative Learngene approach that enhances the training and storage efficiency of descendant models through Repeatable Low-bit Learngene(RELL).
- We redesign the Learngene methodology from perspectives of parameter repetition and quantization, to our knowledge, has not been explored in Learngene the literature.
- Extensive experiments across various datasets demonstrate that our approach achieves superior parameter efficiency and better performance compared to existing methods.

## 2 RELATED WORK

**Learngene.** Learngene (Wang et al., 2023; Shi et al., 2024; Xia et al., 2024d; Feng et al., 2025) is a two-stage framework: first compressing knowledge from a large, well-trained ancestor model into a compact yet generalizable module called learngene, then expanding this module to initialize descendant models of varying scales. HeLG (Wang et al., 2022) extracts some layers from the ancestor model based on gradient information as learngene, and combines them with randomly initialized layers to form a descendant model. PEG (Wang et al., 2024a) utilizes probabilistic sampling for learngene extraction and extends learngene through nonlinear mapping. TLEG (Xia et al., 2024c) and LDR (Lin et al., 2024) initialize descendant models of varying depths through linear combinations of selected learngene modules. However, these methods share a critical limitation: their descendant models maintain full-precision parameters and require complete parameter training, failing to fully realize Learngene's potential for enhancing downstream training efficiency.

**Weight Sharing.** Weight sharing is a parameter-efficient model compression strategy in large pretrained Transformers (Dabre & Fujita, 2019; Lan et al., 2019; Zhang et al., 2022; Takase & Kiyono, 2023). This approach reduces model size by repeat the same parameter set across multiple layers while maintaining comparable model performance. Different from these studies, we propose obtaining repeatable modules and expanding them into descendant models of varying depths through weight sharing, while innovatively integrating weight sharing with parameter-efficient fine-tuning methods.

**Quantization.** Quantization is one of the most effective methods for neural network compression (Banner et al., 2019; Liu et al., 2021). Through quantization-aware training (QAT) (Esser et al., 2020; Li et al., 2022), models can be quantized to lower bit-widths (e.g., 4-bit) without significant accuracy degradation. Notably, recent studies (AskariHemmat et al., 2022; Zhang et al., 2023) have revealed that QAT not only compresses models but also induces a beneficial regularization effect that enhances generalization capability. However, within the Learngene research domain, no existing work has yet explored this intrinsic compatibility between quantization's dual benefits and the Learngene framework's fundamental characteristics.

**Low-rank adapters for fine-tuning.** Low-Rank Adaptation (LoRA) (Hu et al., 2022) is a parameter-efficient fine-tuning technique method. This approach freezes the pre-trained model weights and only trains small low-rank matrices (called adapters). While existing studies (Dettmers et al., 2023; Xu et al., 2024; Li et al., 2024) have investigated combining LoRA with quantization, these methods require importing and freezing the entire model while lacking depth adjustability. In contrast, our approach only needs to incorporate and freeze the learngene (containing merely a subset of layers) while enabling flexible depth adaptation as needed.

## 3 METHODOLOGY

We propose to compress the knowledge of the ancestor model into several low-bit blocks, which can be extended of varying depths. We call these blocks **RELL**(Repeatable Low-bit Learngene). Fig.2 illustrates the entire pipeline of our proposed method, which consists of two steps: Extracting RELL (Step 1) and Expanding RELL (Step 2). We will separately introduce these two steps.

### 3.1 EXTRACTING RELL

In this stage, our objective is to compress the ancestor model's knowledge into several low-bit blocks that capture shared cross-layer knowledge, enabling expansion into descendant models with varying

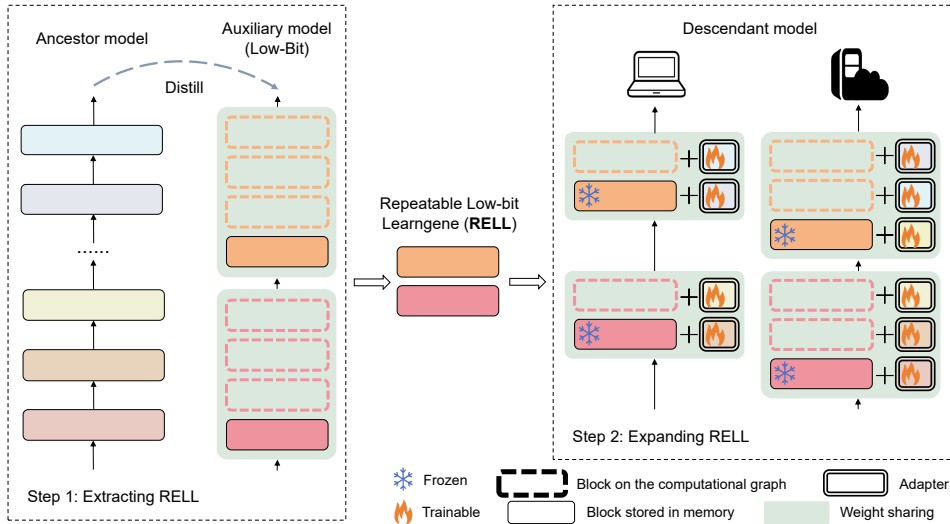

Figure 2: In the first step, we constructed an auxiliary model composed of multiple repeated low-bit blocks. These blocks were then trained to inherit knowledge from the ancestor model through knowledge distillation. Upon completion of training, these low-bit blocks became our repeatable low-bit learngene, termed **RELL**. In the second step, the RELL blocks can be replicated different numbers of times based on downstream task requirements. Their parameters remain frozen, while we introduce a corresponding number of LoRA and fine-tune them to adapt to the specific downstream task.

depths. We first employ a weight-sharing strategy to construct an auxiliary model where these blocks are repeatedly shared across multiple layers, thereby forcing them to learn inter-layer commonalities. Subsequently, we use knowledge distillation to compress the ancestor model's knowledge into these shared low-bit blocks, while employing quantization-aware training(QAT) during optimization. These low-bit blocks ultimately serve as our Repeatable Low-bit Learngene (RELL).

### 3.1.1 AUXILIARY MODEL.

Our auxiliary model follows a framework similar to MiniViT (Zhang et al., 2022), where every $k$ adjacent layers of the model share corresponding transformer module weights for all parameters of multi-head self-attention and multi-layer perceptron, except for LayerNorm. In addition, we have added patch projections and task specific headers to make the auxiliary model suitable for training. However, unlike Minivit, our goal is not to make the auxiliary model perform better, but to use the auxiliary model to obtain the learngene layer, so we did not add a transformation module like Minivit.

$$Z^{i+1} = f(Z^i, MSA_a^i, MLP_a^i, \gamma^i), \ i = 0, 1, ..., L-1 \tag{1}$$

$$MSA_a^i = MSA_R^{\lfloor i/k \rfloor}, \ i = 0, 1, ..., L-1 \tag{2}$$

$$MLP_a^i = MLP_R^{\lfloor i/k \rfloor}, \ i = 0, 1, ..., L-1 \tag{3}$$

where $MSA_R^i$ and $MLP_R^i$ are the parameters of MSA and MLP of RELL in layer $i$, $MSA_a^i$ and $MLP_a^i$ are the parameters of MSA and MLP of auxiliary model in layer $i$, respectively. $\gamma^i$ is the parameters of LN of auxiliary model in layer $i$, which is not shared across different layers of the auxiliary model. $Z_i$ denotes the feature embedding of the sequence in layer $i$ and $L$ is the total number of auxiliary model layers. $\lfloor \cdot \rfloor$ represents rounding downwards. For example, consider an 8-layer ViT auxiliary model where parameters are shared across every 4 layers (as illustrated in the left portion of Fig.2). In this configuration, the RELL would consist of 2 distinct blocks.

### 3.1.2 LEARNING STRATEGY.

While training the auxiliary model to learn RELL's parameters, we employ knowledge distillation to enable RELL to inherit knowledge from the ancestor model. The loss function $\mathcal{L}$ will consist of two parts: classify cross entropy loss $\mathcal{L}_c$ and distillation loss $\mathcal{L}_d$:

$$\mathcal{L}_c = CE(\phi(z_s), y) \tag{4}$$
$$\mathcal{L}_d = KL(\phi(z_s/\tau), \phi(z_t/\tau)) \tag{5}$$
$$\mathcal{L} = \mathcal{L}_c + \mathcal{L}_d \tag{6}$$

where $z_s$ and $z_t$ are the logits output of the auxiliary model and pretrained ancestor model, $y$ denotes ground-truth label, $\phi(\cdot)$ means the softmax function and $CE(\cdot, \cdot)$ means cross entropy loss function. $\tau$ means the temperature for distillation. $KL(\cdot, \cdot)$ represents the Kullback-Leibler divergence loss. To obtain low-bit RELL, we employ Quantization-Aware Training (QAT) during the auxiliary model's training phase to derive their low-bit representations. The specific details of the QAT procedure are provided in the AppendixA.1.

## 3.2 EXPANDING RELL

In this step, our objective is to leverage the RELL obtained in the previous phase to generate descendant models of varying depths, catering to diverse deployment scenarios. To achieve this, we construct low-bit VIT of different depths through parameter repetition, followed by fine-tuning using the QLoRA method.

### 3.2.1 PARAMETER REPETITION.

Firstly, we replicate each RELL layer $r$ times in the computational graph. This means all layers in the descendant models reuse parameters from their corresponding RELL layers. This approach achieves two key benefits: (1) it drastically reduces the model's storage requirements, and (2) enables generation of backbone components for descendant models with varying depths by simply adjusting the value of $r$.

$$MSA_d^i = MSA_R^{\lfloor i/r \rfloor}, \; i = 0, 1, ..., L-1 \tag{7}$$
$$MLP_d^i = MLP_R^{\lfloor i/r \rfloor}, \; i = 0, 1, ..., L-1 \tag{8}$$

where $MSA_d^i$ and $MLP_d^i$ are the parameters of MSA and MLP of descendant model in layer $i$. It should be noted that LN layers are not shared between blocks, and they will be initialized using the parameters trained in the previous step.

### 3.2.2 EFFICIENT PARAMETER FINE-TUNING.

After hierarchically repeating the RELL, we introduced LoRA for each layer of the descendant model. Additionally, we incorporate a patch projection module and task-specific heads, where the patch projection is initialized using parameters obtained from previous training stages and the task-specific heads are randomly initialized. During downstream task adaptation of descendant models, we freeze both the low-bit learngene layers and patch projection module, exclusively fine-tuning all adapters and task-specific heads on downstream tasks.

$$Z^{i+1} = f(Z^i, MSA_d^i, MLP_d^i, \gamma^i, LoRA^i), \; i = 0, 1, ..., L-1 \tag{9}$$

where $LoRA^i$ is the parameters of LoRA of descendant model in layer $i$. This method can generate and fine-tune multiple descendant models of varying depths. For example, as illustrated in the right portion of Fig.2, given a 2-layer RELL, we can repeat each layer twice with corresponding LoRA for each repeat operation, ultimately yielding a 4-layer descendant model. Alternatively, repeating each original layer three times would produce a 6-layer variant.

# 4 EXPERIMENTS

## 4.1 IMPLEMENTATION SETTING

### 4.1.1 DATASET.

To obtain RELL, we first train the auxiliary model on ImageNet-1K (Deng et al., 2009). Subsequently, we train descendant models extended from RELL on several downstream datasets, including iNaturalist 2019(INat-19) (Zhou et al., 2020), CIFAR10(C-10), CIFAR100(C-100) (Krizhevsky et al., 2009), Food-101(F-101) (Bossard et al., 2014), CUB-200-2011(CUB-200) (Wah et al., 2011).

### 4.1.2 ARCHITECTURES.

Following prior work (Xia et al., 2024c), we select LeViT-384 (Graham et al., 2021) as our ancestor model and employ it as the teacher model to transfer knowledge to the auxiliary model through knowledge distillation. Both our auxiliary and descendant models adopt the DeiT-Base (Touvron et al., 2021) architecture, with modifications including layer count adjustment and quantization of internal blocks to 4-bit precision. The auxiliary model consists of three low-bit blocks, with each block shared four times. After training, we extract these three low-bit blocks as our RELL (Repeatable Efficient Low-bit Layers). The descendant models are then constructed by expanding RELL and incorporating LoRA. We design three variants of descendant models with different depths: 6-layer (RELL repeated twice), 9-layer (RELL repeated three times), and 12-layer (RELL repeated four times). Additionally, we also conducted experiments exploring other precision levels and other RELL block counts, with detailed results provided in ablation studies.

### 4.1.3 TRAINING DETAILS.

During the RELL extraction phase, we train the auxiliary model for 150 epochs. Subsequently, we fine-tune the descendant models on downstream tasks for 100 epochs, including a 5-epoch warm-up period. For all tasks, we set the batch size to 128, the initial learning rate to 5e-4, and apply a weight decay of 0.05. Unless otherwise specified, we set the rank of LoRA to 32 across all experiments.

## 4.2 MAIN RESULTS AND ANALYSIS

Following previous Learngene work (Xia et al., 2024c), we first compared RELL against both randomly initialized model and the existing Learngene approach. Additionally, we evaluated RELL's performance against standard LoRA fine-tuning methods.

### 4.2.1 RELL CAN BE EXPANDED TO GENERATE DESCENDANT MODELS OF VARYING DEPTHS.

We conducted experiments on descendant models of varying depths(6, 9, and 12 layers) across downstream datasets. As baselines, we used full-precision models of corresponding depths with random initialization trained from scratch. Fig.3 presents the performance of these descendant models across different downstream tasks. As illustrated, these descendant models achieve superior performance and faster training efficiency. A particularly noteworthy observation is that the performance advantage of RELL-extended models becomes increasingly pronounced as the training dataset size decreases. Taking the 9-layer descendant model as an example. On the INat-19 dataset, it surpasses the baseline's 100-epoch performance in merely 20 training epochs and ultimately achieves a 25% higher accuracy than the fully-trained baseline model. The advantages are even more pronounced on the CUB-200 dataset with limited training samples, where the model exceeds the baseline's final accuracy after just 10 epochs of training and ultimately reaches three times the baseline's classification accuracy. This phenomenon stems from the fact that DeiT models typically require substantial training data, thereby highlighting the crucial importance of leveraging inherited knowledge through our Learngene approach. In addition to cross-dataset transfer evaluations, we further verify RELL's scalability on the source ImageNet-1K dataset, with complete results documented in AppendixA.2.

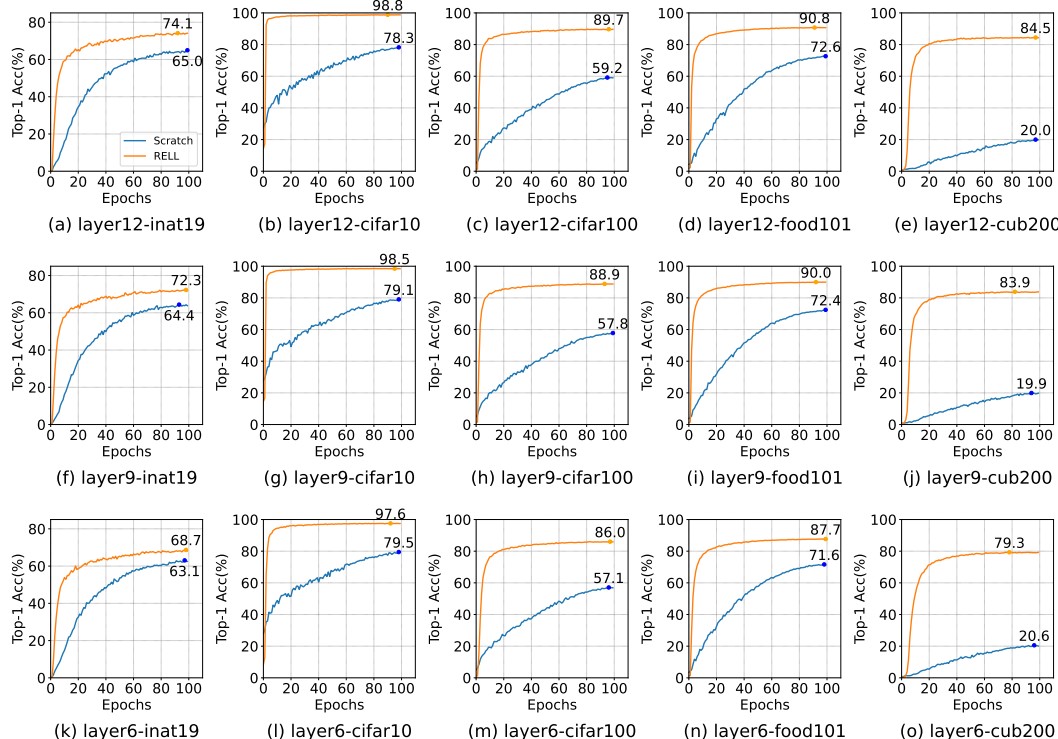

Figure 3: Performance of descendant models with different depths (6, 9, and 12 layers) across various downstream datasets. The orange line represents descendant models expanded using RELL, while the blue line indicates randomly initialized models.

Table 1: Comparisons to existing Learngene approaches. Here, Des-Size (MB) represents the storage size of the descendant model, while TP-Num (M) indicates the number of trainable parameters (in millions) during fine-tuning.

| method | Des-Size(MB) | TP-Num(M) | INat-19 | C-10 | C-100 | F-101 | CUB-200 |
|---|---|---|---|---|---|---|---|
| TLEG | 21.6 | 11.3 | 66.7 | 97.7 | 83.6 | 87.3 | - |
| WAVE | 21.6 | 11.3 | 67.6 | 97.4 | 83.2 | 85.5 | 78.1 |
| PEG | 21.6 | 11.3 | 67.7 | 97.4 | 83.6 | 87.2 | 78.2 |
| Cluster-LG | 21.6 | 11.3 | 71.1 | 97.8 | 85.4 | 89.5 | 76.0 |
| SWS | 42.2 | 22.1 | - | 98.3 | 87.4 | 89.2 | - |
| LETS | 42.2 | 22.1 | - | 98.8 | 89.2 | 90.8 | - |
| RELL(ours) | **20.7** | **4.8** | **74.1** | **98.8** | **89.7** | **90.8** | **84.5** |

### 4.2.2 RELL DEMONSTRATES SUPERIOR PARAMETER EFFICIENCY AND BETTER PERFORMANCE COMPARED TO EXISTING LEARNGENE APPROACHES.

We use a 12 layer descendant model extended with RELL and compare it with other current Learngene methods. These methods include (1) TLEG (Xia et al., 2024c), (2) WAVE (Feng et al., 2025), (3) PEG (Wang et al., 2024a), (4) Cluster-LG (Wang et al., 2024b), (5) SWS (Xia et al., 2024b), and (6) LETS (Xia et al., 2024d). As shown in Table 1, RELL achieves superior performance while maintaining significantly better parameter efficiency and model compactness. For example, on CIFAR100, RELL outperforms the best-performing competitor Cluster-LG (85.4% accuracy) by 4.3 percentage points while using less than half the trainable parameters (4.8M vs 11.3M). Even when compared to larger models, our approach maintains its edge - the 42.2MB LETS model achieves 89.2% accuracy, while RELL delivers comparable performance (89.7%) with less than half the

Table 2: Comparison between training downstream models using RELL and standard LoRA fine-tuning(without weight repetition) under different scenarios.

| scenario | method | Des-Size(MB) | TP-Num(M) | INat-19 | C-10 | C-100 | F-101 | CUB-200 |
|---|---|---|---|---|---|---|---|---|
| (1) | LoRA | 26.4 | 2.4 | 70.5 | 98.2 | 87.6 | 89.0 | 82.7 |
| | RELL | **15.1** | **1.8** | **70.7** | **98.5** | **88.5** | **89.6** | **83.9** |
| (2) | LoRA | 15.6 | 2.4 | 70.1 | 98.4 | 88.2 | 89.2 | 83.3 |
| | RELL | **15.1** | **1.8** | **70.7** | **98.5** | **88.5** | **89.6** | **83.9** |
| (3) | LoRA | 31.1 | 4.8 | 72.9 | 98.8 | 89.6 | 90.6 | **85.2** |
| | RELL | **20.7** | **4.8** | **74.1** | 98.8 | **89.7** | **90.8** | 84.5 |

Table 3: Compared to extracting partial layers from a low-precision DeiT-base.

| method | extracted layers | INat-19 | C-10 | C-100 | F-101 | CUB-200 |
|---|---|---|---|---|---|---|
| Initial | 1, 2, 3 | 64.2 | 92.4 | 72.5 | 77.2 | 40.5 |
| Middle | 6, 7, 8 | 53.5 | 87.4 | 67.5 | 60.6 | 21.1 |
| Final | 10, 11, 12 | 40.7 | 75.1 | 47.0 | 60.6 | 11.3 |
| Distributed | 1, 7, 12 | 63.2 | 92.6 | 73.1 | 74.4 | 50.9 |
| RELL(ours) | - | **74.1** | **98.8** | **89.7** | **90.8** | **84.5** |

model size and only one-fifth the trainable parameters. These results confirm our approach achieves higher parameter efficiency and better performance.

### 4.2.3 RELL OFFERS A MORE COST-EFFECTIVE AND EFFICIENT SOLUTION WHEN RESOURCES ARE LIMITED.

When resources are limited, smaller models are generally chosen for LoRA fine-tuning, which have fewer layers, reduced dimensions, or lower bit-widths. To demonstrate the superiority of our method over standard LoRA(without weight repetition) fine-tuning, we designed the following three comparative scenarios: (1) Reducing model layers: We compared a 9-layer descendant model extended with RELL against a 6-layer 4-bit DeiT-Base fine-tuned with LoRA; (2) Reducing model dimensions: We compared the same 9-layer RELL-extended model against a 12-layer 4-bit DeiT-Small fine-tuned with LoRA; (3) Reducing bit-width: We compared a 12-layer RELL(4bit)-extended model against a 12-layer 2-bit DeiT-Base fine-tuned with LoRA. To ensure a fair comparison, we set the LoRA rank to 16 for scenarios (1) and (2) during RELL extension while keeping it at 32 for other experiments, guaranteeing that the number of trainable parameters in RELL extension remains lower than in LoRA. As shown in Table 2, our method achieves competitive performance with LoRA while using smaller models and fewer training parameters. This indicates that when fine-tuning resources are constrained, employing our RELL approach is a more cost-effective solution compared to reducing model layers, dimensions, or bit-widths.

### 4.3 ABLATION STUDIES

#### 4.3.1 COMPARED TO EXTRACTING PARTIAL LAYERS FROM A LOW-PRECISION DEIT-BASE.

We conducted comparative experiments using a 12-layer descendant model extended from RELL against three-layer extractions from a 12-layer 4-bit DeiT-base model (trained with QAT method under identical settings as RELL). Four extraction approaches were evaluated: (1) initial three layers (layers 1-3), (2) middle three layers (layers 6-8), (3) final three layers (layers 10-12), and (4) distributed layers (layers 1,7,12). All extracted layers were expanded using the same methodology as our RELL extension for downstream tasks. As demonstrated in Table 3, all extraction methods yielded significantly inferior accuracy compared to our approach, particularly in data-scarce scenarios. The performance gap clearly demonstrates that simply extracting arbitrary layers from the model cannot match the efficacy of our RELL extraction method.

Table 4: The performance of descendant models expanded with RELL of varying precision.

| Precision | Des-Size(MB) | INat-19 | C-10 | C-100 | F-101 | CUB-200 |
|---|---|---|---|---|---|---|
| full-precision | 51.1 | **74.7** | 98.6 | 89.8 | 90.7 | 84.8 |
| 8bit | 30.9 | 74.2 | 98.7 | **90.2** | **91.0** | 84.8 |
| 6bit | 25.8 | 73.7 | 98.7 | 89.8 | 90.6 | **85.1** |
| 4bit | 20.7 | 74.1 | **98.8** | 89.7 | 90.8 | 84.5 |
| 2bit | 15.7 | 71.0 | 98.4 | 88.3 | 89.7 | 83.0 |

Table 5: The performance of descendant models expanded with RELL of varying block number.

| Block-Num | Des-Size(MB) | INat-19 | C-10 | C-100 | F-101 | CUB-200 |
|---|---|---|---|---|---|---|
| 1 | 13.9 | 71.3 | 98.3 | 87.5 | 89.3 | 82.3 |
| 2 | 17.3 | 73.7 | 98.6 | 89.4 | 90.2 | 84.4 |
| 3 | 20.7 | 74.1 | 98.8 | 89.7 | 90.8 | 84.5 |
| 4 | 24.2 | 73.4 | 98.8 | 90.2 | 91.0 | 85.2 |
| 6 | 31.0 | 73.8 | 98.8 | 90.4 | 91.1 | 85.7 |

### 4.3.2 THE IMPACT OF RELLs WITH DIFFERENT PRECISION LEVELS.

During quantization-aware training of the auxiliary model, we set the precision of its shared blocks to 2-bit, 4-bit, 6-bit, 8-bit and full-precision, thereby obtaining RELLs of varying precision levels. Table 4 presents the performance of the resulting 12-layer descendant models expanded from these RELLs. As shown, on most datasets, descendant models expanded from low-bit RELLs outperform those expanded from full-precision RELLs. This indicates that quantization not only reduces model size but also contributes to improved generalization. However, when the RELL precision is further reduced (e.g., to 2-bit), the performance of the descendant models falls below that of models expanded from full-precision RELLs. This suggests that the loss of accuracy caused by excessively low precision outweighs the benefits gained from generalization.

### 4.3.3 THE IMPACT OF RELLs WITH DIFFERENT NUMBER OF BLOCKS.

We maintained the auxiliary model at a fixed depth of 12 layers, and shared the blocks within the auxiliary model 12, 6, 4, 3, and 2 times respectively—meaning that 1, 2, 3, 4, and 6 RELL blocks were extracted from the ancestor model. We then expanded these into 12-layer descendant models on downstream datasets. As presented in Table 5, the results demonstrate a clear trade-off: while model performance improves with increasing numbers of RELL blocks, this enhancement comes at the cost of larger model sizes. Additionally, we explored the scenario where three RELL blocks were extracted from the ancestor model, and investigated the performance of descendant models extended using one, two, or all three of these RELL blocks. Detailed results are provided in AppendixA.3.

## 5 CONCLUSION

In conclusion, we propose an innovative Learngene approach, redesigning the Learngene methodology from perspectives of parameter repetition and quantization. Our approach compresses the ancestor model's knowledge into low-bit repeatable blocks termed RELL (Repeatable Low-bit Learngene), which can be jointly extended with LoRA fine-tuning to construct Vision Transformers of varying depths for different downstream tasks. Experimental results demonstrate the superior effectiveness and flexibility of our method compared to existing approaches. This solution provides a computationally economical framework for building ViTs tailored to diverse requirements and application scenarios.

## ETHICS STATEMENT

This work adheres to the ICLR Code of Ethics. Our research is based on the publicly available dataset. We did not collect new data nor involve human subjects directly.

## REPRODUCIBILITY STATEMENT

All experiments in this article were conducted on NVIDIA RTX 3090 GPU servers. Hyperparameter settings are detailed in Section 4.1.3. The code for the experiments has been uploaded in the supplementary materials.

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

# A APPENDIX

## A.1 THE DETAILS OF THE QAT PROCEDURE

Our method maintains full accuracy for activation and only apply symmetric uniform quantization to the model weights. We use the Quantization Aware Training (QAT) method to learn the low bit weights and corresponding quantization parameters of the model. During training, the forward propagation of each linear layer consists of two steps: quantization and de-quantization. For the full precision weight $w$ of a certain linear layer and the $b$ bit width of quantization, the quantization operation is as follows:

$$\hat{w} = \lfloor clip\{w/\alpha_w, Q_n, Q_p\} \rceil \tag{10}$$

$$Q_n = -2^{b-1} \tag{11}$$

$$Q_p = 2^{b-1} - 1 \tag{12}$$

where $\alpha_w$ is a trainable scaling factor, and $Q_n$ and $Q_p$ represent the minimum and maximum values of the quantized weight $\hat{w}$. Respectively, $clip\{\cdot, min, max\}$ clips data that exceeds the minimum or maximum range, and $\lfloor \cdot \rceil$ rounds data to the nearest integer. This quantization operation maps $w$ to a discrete value in $\{-2^{b-1}, ..., -1, 0, 1, ..., 2^{b-1} - 1\}$. Then, $\hat{w}$ will be performed de-quantization to output the quantized weight $\bar{w}$ through the following operation:

$$\bar{w} = \hat{w} \times \alpha_w \tag{13}$$

The forward propagation process of the linear layer during QAT is as follows:

$$y = linear(x, \bar{w}) \tag{14}$$

where $x$ is the full precision input of the linear layer and $y$ is the output. In addition, we use the straight through estimator(STE) (Bengio et al., 2013) during training to approximate the gradient of the rounding operator as 1.

$$\frac{\partial \lfloor x \rceil}{\partial x} = 1 \tag{15}$$

It should be noted that we quantize only the model's block components while maintaining full precision for patch projection and header parts.

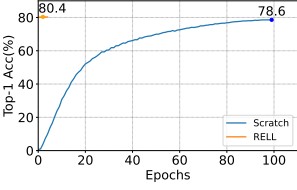 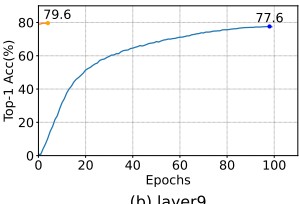 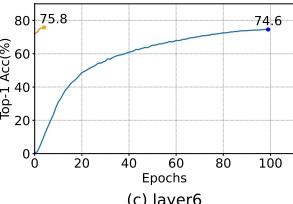

(a) layer12                          (b) layer9                          (c) layer6

Figure 4: The performance of depth-variable descendant models expanded from 4-bit RELL of 3 blocks on ImageNet-1k.

Table 6: In the case where three RELL blocks were extracted from the ancestor model, the performance of the descendant models expanded using one, two, or all three of these blocks.

| Block-Num | INat-19 | C-10 | C-100 | F-101 | CUB-200 |
|---|---|---|---|---|---|
| 1 | 67.2 | 94.4 | 76.4 | 84.9 | 53.5 |
| 2 | 71.9 | 98.0 | 86.7 | 89.9 | 81.0 |
| 3 | **74.1** | **98.8** | **89.7** | **90.8** | **84.5** |

## A.2 THE PERFORMANCE OF DEPTH-VARIABLE DESCENDANT MODELS EXPANDED FROM 4-BIT RELL OF 3 BLOCKS ON IMAGENET-1K.

We conducted experiments on RELL(4bit)-extended descendant models of varying depths(6, 9, and 12 layers) on ImageNet-1K. As baselines, we used full-precision models of corresponding depths with random initialization trained from scratch. Fig.4 presents the performance of these descendant models on the ImageNet-1K classification task. As illustrated in Fig.4, descendant models of varying depths extended using RELL demonstrate both higher accuracy and faster convergence rates. Notably, the 4-bit RELL-extended descendant models achieve comparable performance to models trained from scratch for 100 epochs, while requiring only 5 training epochs. These results strongly indicate that RELL possesses excellent scalability across different model depths.

## A.3 THE IMPACT OF VARYING RELL BLOCK SELECTION FROM A SET OF THREE ON DESCENDANT MODEL PERFORMANCE.

Under the condition of extracting three RELL blocks from the ancestor model, we conducted experiments constructing 12-layer descendant models using one, two, or all three of these blocks, respectively. As shown in Table 6, we observe that the performance of descendant models on downstream datasets progressively improves as more RELL blocks are incorporated. This clearly demonstrates the effectiveness of each individual block within the RELL.

## A.4 THE USE OF LARGE LANGUAGE MODELS (LLMS)

Large Language Models (LLMs) were used solely for article polishing and grammatical correction. Specifically, the LLM assisted in (i) refining grammar and improving fluency, and (ii) standardizing terminology, tense, and voice. The LLM was not involved in designing experiments, analyzing data, or drawing conclusions. All methodologies, experimental designs, and findings were authored, validated, and interpreted by the human authors. We thoroughly reviewed all LLM-assisted edits to ensure accuracy and adherence to the intended meaning.

